# What do you learn from context? Probing for sentence structure in contextualized word representations

**Ian Tenney,**[*][1] **Patrick Xia,**[2] **Berlin Chen,**[3] **Alex Wang,**[4] **Adam Poliak,**[2]
**R. Thomas McCoy,**[2] **Najoung Kim,**[2] **Benjamin Van Durme,**[2] **Samuel R. Bowman,**[4]
**Dipanjan Das,**[1] **and Ellie Pavlick**[1,5]

[1]Google AI Language, [2]Johns Hopkins University, [3]Swarthmore College,
[4]New York University, [5]Brown University

## Abstract

Contextualized representation models such as ELMo (Peters et al., 2018a) and BERT (Devlin et al., 2018) have recently achieved state-of-the-art results on a diverse array of downstream NLP tasks. Building on recent token-level probing work, we introduce a novel *edge probing* task design and construct a broad suite of sub-sentence tasks derived from the traditional structured NLP pipeline. We probe word-level contextual representations from four recent models and investigate how they encode sentence structure across a range of syntactic, semantic, local, and long-range phenomena. We find that existing models trained on language modeling and translation produce strong representations for syntactic phenomena, but only offer comparably small improvements on semantic tasks over a non-contextual baseline.

## 1 Introduction[1]

Pretrained word embeddings (Mikolov et al., 2013; Pennington et al., 2014) are a staple tool for NLP. These models provide continuous representations for word types, typically learned from co-occurrence statistics on unlabeled data, and improve generalization of downstream models across many domains. Recently, a number of models have been proposed for *contextualized* word embeddings. Instead of using a single, fixed vector per word type, these models run a pretrained encoder network over the sentence to produce contextual embeddings of each token. The encoder, usually an LSTM (Hochreiter & Schmidhuber, 1997) or a Transformer (Vaswani et al., 2017), can be trained on objectives like machine translation (McCann et al., 2017) or language modeling (Peters et al., 2018a; Radford et al., 2018; Howard & Ruder, 2018; Devlin et al., 2018), for which large amounts of data are available. The activations of this network–a collection of one vector per token–fit the same interface as conventional word embeddings, and can be used as a drop-in replacement input to any model. Applied to popular models, this technique has yielded significant improvements to the state-of-the-art on several tasks, including constituency parsing (Kitaev & Klein, 2018), semantic role labeling (He et al., 2018; Strubell et al., 2018), and coreference (Lee et al., 2018), and has outperformed competing techniques (Kiros et al., 2015; Conneau et al., 2017) that produce fixed-length representations for entire sentences.

Our goal in this work is to understand where these contextual representations improve over conventional word embeddings. Recent work has explored many token-level properties of these representations, such as their ability to capture part-of-speech tags (Blevins et al., 2018; Belinkov et al., 2017b; Shi et al., 2016), morphology (Belinkov et al., 2017a;b), or word-sense disambiguation (Peters et al., 2018a). Peters et al. (2018b) extends this to constituent phrases, and present a heuristic for unsuper-

---

[*]Correspondence: `iftenney@google.com`. This work was partly conducted at the 2018 JSALT workshop at Johns Hopkins University.

[1]This paper has been updated from the original version, primarily to include results on BERT (Devlin et al., 2018). See Appendix A for a detailed list of changes.

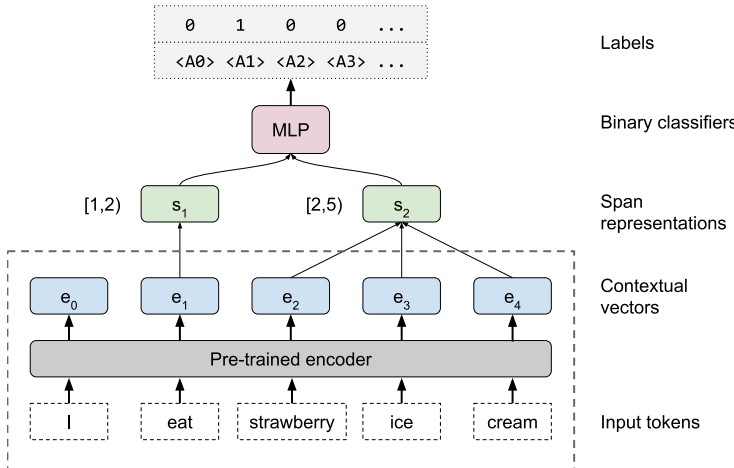

Figure 1: Probing model architecture (§ 3.1). All parameters inside the dashed line are fixed, while we train the span pooling and MLP classifiers to extract information from the contextual vectors. The example shown is for semantic role labeling, where $s^{(1)} = [1, 2)$ corresponds to the predicate ("eat"), while $s^{(2)} = [2, 5)$ is the argument ("strawberry ice cream"), and we predict label $\texttt{A1}$ as positive and others as negative. For entity and constituent labeling, only a single span is used.

vised pronominal coreference. We expand on this even further and introduce a suite of *edge probing* tasks covering a broad range of syntactic, semantic, local, and long-range phenomena. In particular, we focus on asking what information is encoded at each position, and how well it encodes structural information about that word's role in the sentence. Is this information primarily syntactic in nature, or do the representations also encode higher-level semantic relationships? Is this information local, or do the encoders also capture long-range structure?

We approach these questions with a probing model (Figure 1) that sees only the contextual embeddings from a fixed, pretrained encoder. The model can access only embeddings within given spans, such as a predicate-argument pair, and must predict properties, such as semantic roles, which typically require whole-sentence context. We use data derived from traditional structured NLP tasks: tagging, parsing, semantic roles, and coreference. Common corpora such as OntoNotes (Weischedel et al., 2013) provide a wealth of annotations for well-studied concepts which are both linguistically motivated and known to be useful intermediates for high-level language understanding. We refer to our technique as "edge probing", as we decompose each structured task into a set of graph edges (§ 2) which we can predict independently using a common classifier architecture (§ 3.1)[2]. We probe four popular contextual representation models (§ 3.2): CoVe (McCann et al., 2017), ELMo (Peters et al., 2018a), OpenAI GPT (Radford et al., 2018), and BERT (Devlin et al., 2018).

We focus on these models because their pretrained weights and code are available, since these are most likely to be used by researchers. We compare to word-level baselines to separate the contribution of context from lexical priors, and experiment with augmented baselines to better understand the role of pretraining and the ability of encoders to capture long-range dependencies.

## 2 EDGE PROBING

To carry out our experiments, we define a novel "edge probing" framework motivated by the need for a uniform set of metrics and architectures across tasks. Our framework is generic, and can be applied to any task that can be represented as a labeled graph anchored to spans in a sentence.

**Formulation.** Formally, we represent a sentence as a list of tokens $T = [t_0, t_1, \ldots, t_n]$, and a labeled edge as $\{s^{(1)}, s^{(2)}, L\}$. We treat $s^{(1)} = [i^{(1)}, j^{(1)})$ and, optionally, $s^{(2)} = [i^{(2)}, j^{(2)})$ as (end-exclusive) spans. For unary edges such as constituent labels, $s^{(2)}$ is omitted. We take $L$ to be a set of zero or more targets from a task-specific label set $\mathcal{L}$.

---

[2]Our code is publicly available at $\texttt{https://github.com/jsalt18-sentence-repl/jiant}$.

| | |
|---|---|
| POS | The important thing about Disney is that it is a global [brand]$_1$. $\rightarrow$ NN (Noun) |
| Constit. | The important thing about Disney is that it [is a global brand]$_1$. $\rightarrow$ VP (Verb Phrase) |
| Depend. | [Atmosphere]$_1$ is always [fun]$_2$ $\rightarrow$ nsubj (nominal subject) |
| Entities | The important thing about [Disney]$_1$ is that it is a global brand. $\rightarrow$ Organization |
| SRL | [The important thing about Disney]$_2$ [is]$_1$ that it is a global brand. $\rightarrow$ Arg1 (Agent) |
| SPR | [It]$_1$ [endorsed]$_2$ the White House strategy... $\rightarrow$ {awareness, existed_after, ...} |
| Coref.$^{\text{O}}$ | The important thing about [Disney]$_1$ is that [it]$_2$ is a global brand. $\rightarrow$ True |
| Coref.$^{\text{W}}$ | [Characters]$_2$ entertain audiences because [they]$_1$ want people to be happy. $\rightarrow$ True Characters entertain [audiences]$_2$ because [they]$_1$ want people to be happy. $\rightarrow$ False |
| Rel. | The [burst]$_1$ has been caused by water hammer [pressure]$_2$. $\rightarrow$ Cause-Effect($e_2$, $e_1$) |

Table 1: Example sentence, spans, and target label for each task. O = OntoNotes, W = Winograd.

To cast all tasks into a common classification model, we focus on the *labeling* versions of each task. Spans (gold mentions, constituents, predicates, etc.) are given as inputs, and the model is trained to predict $L$ as a multi-label target. We note that this is only one component of the common pipelined (or end-to-end) approach to these tasks, and that in general our metrics are not comparable to models that jointly perform span *identification* and labeling. However, since our focus is on analysis rather than application, the labeling version is a better fit for our goals of isolating individual phenomena of interest, and giving a uniform metric – binary F1 score – across our probing suite.

## 2.1 TASKS

Our experiments focus on eight core NLP labeling tasks: part-of-speech, constituents, dependencies, named entities, semantic roles, coreference, semantic proto-roles, and relation classification. The tasks and their respective datasets are described below, and also detailed in Table 1 and Appendix B.

**Part-of-speech tagging (POS)** is the syntactic task of assigning tags such as noun, verb, adjective, etc. to individual tokens. We let $s_1 = [i, i + 1)$ be a single token, and seek to predict the POS tag.

**Constituent labeling** is the more general task concerned with assigning a non-terminal label for a span of tokens within the phrase-structure parse of the sentence: e.g. is the span a noun phrase, a verb phrase, etc. We let $s_1 = [i, j)$ be a known constituent, and seek to predict the constituent label.

**Dependency labeling** is similar to constituent labeling, except that rather than aiming to position a span of tokens within the phrase structure, dependency labeling seeks to predict the functional relationships of one token relative to another: e.g. is in a modifier-head relationship, a subject-object relationship, etc. We take $s_1 = [i, i + 1)$ to be a single token and $s_2 = [j, j + 1)$ to be its syntactic head, and seek to predict the dependency relation between tokens $i$ and $j$.

**Named entity labeling** is the task of predicting the category of an entity referred to by a given span, e.g. does the entity refer to a person, a location, an organization, etc. We let $s_1 = [i, j)$ represent an entity span and seek to predict the entity type.

**Semantic role labeling (SRL)** is the task of imposing predicate-argument structure onto a natural language sentence: e.g. given a sentence like *"Mary pushed John"*, SRL is concerned with identifying *"Mary"* as the pusher and *"John"* as the pushee. We let $s_1 = [i_1, j_1)$ represent a known predicate and $s_2 = [i_2, j_2)$ represent a known argument of that predicate, and seek to predict the role that the argument $s_2$ fills–e.g. ARG0 (agent, the *pusher*) vs. ARG1 (patient, the *pushee*).

**Coreference** is the task of determining whether two spans of tokens ("mentions") refer to the same entity (or event): e.g. in a given context, do *"Obama"* and *"the former president"* refer to the same person, or do *"New York City"* and *"there"* refer to the same place. We let $s_1$ and $s_2$ represent known mentions, and seek to make a binary prediction of whether they co-refer.

**Semantic proto-role (SPR)** labeling is the task of annotating fine-grained, non-exclusive semantic attributes, such as change_of_state or awareness, over predicate-argument pairs. E.g.

given the sentence *"Mary pushed John"*, whereas SRL is concerned with identifying *"Mary"* as the pusher, SPR is concerned with identifying attributes such as `awareness` (whether the pusher is *aware* that they are doing the pushing). We let $s_1$ represent a predicate span and $s_2$ a known argument head, and perform a multi-label classification over potential attributes of the predicate-argument relation.

**Relation Classification (Rel.)** is the task of predicting the real-world relation that holds between two entities, typically given an inventory of symbolic relation types (often from an ontology or database schema). For example, given a sentence like *"Mary is walking to work"*, relation classification is concerned with linking *"Mary"* to *"work"* via the `Entity-Destination` relation. We let $s_1$ and $s_2$ represent known mentions, and seek to predict the relation type.

## 2.2 DATASETS

We use the annotations in the OntoNotes 5.0 corpus (Weischedel et al., 2013) for five of the above eight tasks: POS tags, constituents, named entities, semantic roles, and coreference. In all cases, we simply cast the original annotation into our edge probing format. For POS tagging, we simply extract these labels from the constituency parse data in OntoNotes. For coreference, since OntoNotes only provides annotations for positive examples (pairs of mentions that corefer) we generate negative examples by generating all pairs of mentions that are not explicitly marked as coreferent.

The OntoNotes corpus does not contain annotations for dependencies, proto-roles, or semantic relations. Thus, for dependencies, we use the English Web Treebank portion of the Universal Dependencies 2.2 release (Silveira et al., 2014). For SPR, we use two datasets, one (SPR1; Teichert et al. (2017)) derived from Penn Treebank and one (SPR2; Rudinger et al. (2018)) derived from English Web Treebank. For relation classification, we use the SemEval 2010 Task 8 dataset (Hendrickx et al., 2009), which consists of sentences sampled from English web text, labeled with a set of 9 directional relation types.

In addition to the OntoNotes coreference examples, we include an extra "challenge" coreference dataset based on the Winograd schema (Levesque et al., 2012). Winograd schema problems focus on cases of pronoun resolution which are syntactically ambiguous and thus are intended to require subtler semantic inference in order to resolve correctly (see example in Table 1). We use the version of the Definite Pronoun Resolution (DPR) dataset (Rahman & Ng, 2012) employed by White et al. (2017), which contains balanced positive and negative pairs.

## 3 EXPERIMENTAL SET-UP

### 3.1 PROBING MODEL

Our probing architecture is illustrated in Figure 1. The model is designed to have limited expressive power on its own, as to focus on what information can be extracted from the contextual embeddings. We take a list of contextual vectors $[e_0, e_1, \ldots, e_n]$ and integer spans $s^{(1)} = [i^{(1)}, j^{(1)})$ and (optionally) $s^{(2)} = [i^{(2)}, j^{(2)})$ as inputs, and use a projection layer followed by the self-attention pooling operator of Lee et al. (2017) to compute fixed-length span representations. Pooling is only within the bounds of a span, e.g. the vectors $[e_i, e_{i+1}, \ldots, e_{j-1}]$, which means that the only information our model can access about the rest of the sentence is that provided by the contextual embeddings.

The span representations are concatenated and fed into a two-layer MLP followed by a sigmoid output layer. We train by minimizing binary cross-entropy against the target label set $L \in \{0, 1\}^{|\mathcal{L}|}$. Our code is implemented in PyTorch (Paszke et al., 2017) using the AllenNLP (Gardner et al., 2018) toolkit. For further details on training, see Appendix C.

### 3.2 SENTENCE REPRESENTATION MODELS

We explore four recent contextual encoder models: CoVe, ELMo, OpenAI GPT, and BERT. Each model takes tokens $[t_0, t_1, \ldots, t_n]$ as input and produces a list of contextual vectors $[e_0, e_1, \ldots, e_n]$.

**CoVe** (McCann et al., 2017) uses the top-level activations of a two-layer biLSTM trained on English-German translation, concatenated with 300-dimensional GloVe vectors. The source data consists of

7 million sentences from web crawl, news, and government proceedings (WMT 2017; Bojar et al. (2017)).

**ELMo** (Peters et al., 2018a) is a two-layer bidirectional LSTM language model, built over a context-independent character CNN layer and trained on the Billion Word Benchmark dataset (Chelba et al., 2014), consisting primarily of newswire text. We follow standard usage and take a linear combination of the ELMo layers, using learned task-specific scalars (Equation 1 of Peters et al., 2018a).

**GPT** (Radford et al., 2018) is a 12-layer Transformer (Vaswani et al., 2017) encoder trained as a left-to-right language model on the Toronto Books Corpus (Zhu et al., 2015). Departing from the original authors, we do not fine-tune the encoder[3].

**BERT** (Devlin et al., 2018) is a deep Transformer (Vaswani et al., 2017) encoder trained jointly as a masked language model and on next-sentence prediction, trained on the concatenation of the Toronto Books Corpus (Zhu et al., 2015) and English Wikipedia. As with GPT, we do not fine-tune the encoder weights. We probe the publicly released `bert-base-uncased` (12-layer) and `bert-large-uncased` (24-layer) models[4].

For BERT and GPT, we compare two methods for yielding contextual vectors for each token: **`cat`** where we concatenate the subword embeddings with the activations of the top layer, similar to CoVe, and **`mix`** where we take a linear combination of layer activations (including embeddings) using learned task-specific scalars (Equation 1 of Peters et al., 2018a), similar to ELMo.

The resulting contextual vectors have dimension $d = 900$ for CoVe, $d = 1024$ for ELMo, and $d = 1536$ (`cat`) or $d = 768$ (`mix`) for GPT and BERT-base, and $d = 2048$ (`cat`) or $d = 1024$ (`mix`) for BERT-large[5]. The pretrained models expect different tokenizations and input processing. We use a heuristic alignment algorithm based on byte-level Levenshtein distance, explained in detail in Appendix E, in order to re-map spans from the source data to the tokenization expected by the above models.

## 4 EXPERIMENTS

Again, we want to answer: What do contextual representations encode that conventional word embeddings do not? Our experimental comparisons, described below, are intended to ablate various aspects of contextualized encoders in order to illuminate how the model captures different types of linguistic information.

**Lexical Baselines.**   In order to probe the effect of each *contextual* encoder, we train a version of our probing model directly on the most closely related context-independent word representations. This baseline measures the performance that can be achieved from lexical priors alone, without any access to surrounding words. For CoVe, we compare to the embedding layer of that model, which consists of 300-dimensional GloVe vectors trained on 840 billion tokens of CommonCrawl (web) text. For ELMo, we use the activations of the context-independent character-CNN layer (layer 0) from the full model. For GPT and for BERT, we use the learned subword embeddings from the full model.

**Randomized ELMo.**   Randomized neural networks have recently (Zhang & Bowman, 2018) shown surprisingly strong performance on many tasks, suggesting that architecture may play a significant role in learning useful feature functions. To help understand what is actually *learned* during the encoder pretraining, we compare with a version of the ELMo model in which all weights above the lexical layer (layer 0) are replaced with random orthonormal matrices[6].

---

[3]We note that there may be information not easily accessible without fine-tuning the LSTM weights. This can be easily explored within our framework, e.g. using the techniques of Howard & Ruder (2018) or Radford et al. (2018). We leave this to future work, and hope that our code release will facilitate such continuations.

[4]Devlin et al. (2018) recommend the `cased` BERT models for named entity *recognition* tasks; however, we find no difference in performance on our entity *labeling* variant and so report all results with `uncased` models.

[5]For further details, see Appendix D.

[6]This includes both LSTM cell weights and projection matrices between layers. Non-square matrices are orthogonal along the smaller dimension.

**Word-Level CNN.** To what extent do contextual encoders capture long-range dependencies, versus simply modeling local context? We extend our lexical baseline by introducing a fixed-width convolutional layer on top of the word representations. As comparing to the lexical baseline factors out word-level priors, comparing to this CNN baseline factors out local relationships, such as the presence of nearby function words, and allows us to see the contribution of long-range context to encoder performance. To implement this, we replace the projection layer in our probing model with a fully-connected CNN that sees $\pm 1$ or $\pm 2$ tokens around the center word (i.e. kernel width 3 or 5).

## 5 RESULTS

Using the above experimental design, we return to the central questions originally posed. That is, what types of syntactic and semantic information does each model encode at each position? And is the information captured primarily local, or do contextualized embeddings encode information about long-range sentential structure?

**Comparison of representation models.** We report F1 scores for ELMo, CoVe, GPT, and BERT in Table 2. We observe that ELMo and GPT (with `mix` features) have comparable performance, with ELMo slightly better on most tasks but the Transformer scoring higher on relation classification and OntoNotes coreference. Both models outperform CoVe by a significant margin (6.3 F1 points on average), meaning that the information in their word representations makes it easier to recover details of sentence structure. It is important to note that while ELMo, CoVe, and the GPT can be applied to the same problems, they differ in architecture, training objective, and both the quantity and genre of training data (§ 3.2). Furthermore, on all tasks except for Winograd coreference, the lexical representations used by the ELMo and GPT models outperform GloVe vectors (by 5.4 and 2.4 points on average, respectively). This is particularly pronounced on constituent and semantic role labeling, where the model may be benefiting from better handling of morphology by character-level or subword representations.

We observe that using ELMo-style scalar mixing (`mix`) instead of concatenation improves performance significantly (1-3 F1 points on average) on both deep Transformer models (BERT and GPT). We attribute this to the most relevant information being contained in intermediate layers, which agrees with observations by Blevins et al. (2018), Peters et al. (2018a), and Devlin et al. (2018), and with the finding of Peters et al. (2018b) that top layers may be overly specialized to perform next-word prediction.

When using scalar mixing (`mix`), we observe that the BERT-base model outperforms GPT, which has a similar 12-layer Transformer architecture, by approximately 2 F1 points on average. The 24-layer BERT-large model performs better still, besting BERT-base by 1.1 F1 points and ELMo by 2.7 F1 - a nearly 20% relative reduction in error on most tasks.

We find that the improvements of the BERT models are not uniform across tasks. In particular, BERT-large improves on ELMo by 7.4 F1 points on OntoNotes coreference, more than a 40% reduction in error and nearly as high as the improvement of the ELMo encoder over its lexical baseline. We also see a large improvement (7.8 F1 points)[7] on Winograd-style coreference from BERT-large in particular, suggesting that deeper unsupervised models may yield further improvement on difficult semantic tasks.

**Genre Effects.** Our probing suite is drawn mostly from newswire and web text (§ 2). This is a good match for the Billion Word Benchmark (BWB) used to train the ELMo model, but a weaker match for the Books Corpus used to train the published GPT model. To control for this, we train a clone of the GPT model on the BWB, using the code and hyperparameters of Radford et al. (2018). We find that this model performs only slightly better (+0.15 F1 on average) on our probing suite than the Books Corpus-trained model, but still underperforms ELMo by nearly 1 F1 point.

**Encoding of syntactic vs. semantic information.** By comparing to lexical baselines, we can measure how much the contextual information from a particular encoder improves performance on

---

[7]On average; the DPR dataset has high variance and we observe a mix of runs which score in the mid-50s and the high-60s F1.

| | CoVe | | | ELMo | | | GPT | | |
|---|---|---|---|---|---|---|---|---|---|
| | **Lex.** | **Full** | **Abs. Δ** | **Lex.** | **Full** | **Abs. Δ** | **Lex.** | **cat** | **mix** |
| Part-of-Speech | 85.7 | 94.0 | 8.4 | 90.4 | **96.7** | 6.3 | 88.2 | 94.9 | 95.0 |
| Constituents | 56.1 | 81.6 | 25.4 | 69.1 | **84.6** | 15.4 | 65.1 | 81.3 | **84.6** |
| Dependencies | 75.0 | 83.6 | 8.6 | 80.4 | **93.9** | 13.6 | 77.7 | 92.1 | **94.1** |
| Entities | 88.4 | 90.3 | 1.9 | 92.0 | **95.6** | 3.5 | 88.6 | 92.9 | 92.5 |
| SRL (all) | 59.7 | 80.4 | 20.7 | 74.1 | **90.1** | 16.0 | 67.7 | 86.0 | 89.7 |
| *Core roles* | *56.2* | *81.0* | *24.7* | *73.6* | ***92.6*** | *19.0* | *65.1* | *88.0* | *92.0* |
| *Non-core roles* | *67.7* | *78.8* | *11.1* | *75.4* | ***84.1*** | *8.8* | *73.9* | *81.3* | ***84.1*** |
| OntoNotes coref. | 72.9 | 79.2 | 6.3 | 75.3 | 84.0 | 8.7 | 71.8 | 83.6 | **86.3** |
| SPR1 | 73.7 | 77.1 | 3.4 | 80.1 | **84.8** | 4.7 | 79.2 | 83.5 | 83.1 |
| SPR2 | 76.6 | 80.2 | 3.6 | 82.1 | 83.1 | 1.0 | 82.2 | **83.8** | 83.5 |
| Winograd coref. | 52.1 | **54.3** | 2.2 | **54.3** | 53.5 | -0.8 | 51.7 | 52.6 | **53.8** |
| Rel. (SemEval) | 51.0 | 60.6 | 9.6 | 55.7 | 77.8 | 22.1 | 58.2 | **81.3** | 81.0 |
| Macro Average | 69.1 | 78.1 | 9.0 | 75.4 | **84.4** | 9.1 | 73.0 | 83.2 | **84.4** |

| | BERT-base | | | | BERT-large | | | | |
|---|---|---|---|---|---|---|---|---|---|
| | **F1 Score** | | | **Abs. Δ** | **F1 Score** | | | **Abs. Δ** | |
| | **Lex.** | **cat** | **mix** | **ELMo** | **Lex.** | **cat** | **mix** | **(base)** | **ELMo** |
| Part-of-Speech | 88.4 | **97.0** | 96.7 | 0.0 | 88.1 | 96.5 | **96.9** | 0.2 | 0.2 |
| Constituents | 68.4 | 83.7 | 86.7 | 2.1 | 69.0 | 80.1 | **87.0** | 0.4 | 2.5 |
| Dependencies | 80.1 | 93.0 | 95.1 | 1.1 | 80.2 | 91.5 | **95.4** | 0.3 | 1.4 |
| Entities | 90.9 | 96.1 | 96.2 | 0.6 | 91.8 | 96.2 | **96.5** | 0.3 | 0.9 |
| SRL (all) | 75.4 | 89.4 | 91.3 | 1.2 | 76.5 | 88.2 | **92.3** | 1.0 | 2.2 |
| *Core roles* | *74.9* | *91.4* | *93.6* | *1.0* | *76.3* | *89.9* | *94.6* | *1.0* | *2.0* |
| *Non-core roles* | *76.4* | *84.7* | *85.9* | *1.8* | *76.9* | *84.1* | *86.9* | *1.0* | *2.8* |
| OntoNotes coref. | 74.9 | 88.7 | 90.2 | 6.3 | 75.7 | 89.6 | 91.4 | 1.2 | 7.4 |
| SPR1 | 79.2 | 84.7 | **86.1** | 1.3 | 79.6 | 85.1 | 85.8 | -0.3 | 1.0 |
| SPR2 | 81.7 | 83.0 | **83.8** | 0.7 | 81.6 | 83.2 | 84.1 | 0.3 | 1.0 |
| Winograd coref. | 54.3 | 53.6 | 54.9 | 1.4 | 53.0 | 53.8 | 61.4 | 6.5 | 7.8 |
| Rel. (SemEval) | 57.4 | 78.3 | 82.0 | 4.2 | 56.2 | 77.6 | 82.4 | 0.5 | 4.6 |
| Macro Average | 75.1 | 84.8 | 86.3 | 1.9 | 75.2 | 84.2 | **87.3** | 1.0 | 2.9 |

Table 2: Comparison of representation models and their respective lexical baselines. Numbers reported are micro-averaged F1 score on respective test sets. **Lex.** denotes the lexical baseline (§ 4) for each model, and bold denotes the best performance on each task. Lines in *italics* are subsets of the targets from a parent task; these are omitted in the macro average. SRL numbers consider core and non-core roles, but ignore references and continuations. Winograd (DPR) results are the average of five runs each using a random sample (without replacement) of 80% of the training data. 95% confidence intervals (normal approximation) are approximately $\pm 3$ ($\pm 6$ with BERT-large) for Winograd, $\pm 1$ for SPR1 and SPR2, and $\pm 0.5$ or smaller for all other tasks.

each task. Note that in all cases, the contextual representation is strictly more expressive, since it includes access to the lexical representations either by concatenation or by scalar mixing.

We observe that ELMo, CoVe, and GPT all follow a similar trend across our suite (Table 2), showing the largest gains on tasks which are considered to be largely syntactic, such as dependency and constituent labeling, and smaller gains on tasks which are considered to require more semantic reasoning, such as SPR and Winograd. We observe small absolute improvements (+6.3 and +3.5 for ELMo Full vs. Lex.) on part-of-speech tagging and entity labeling, but note that this is likely due to the strength of word-level priors on these tasks. Relative reduction in error is much higher (+66% for Part-of-Speech and +44% for Entities), suggesting that ELMo does encode local type information.

Semantic role labeling benefits greatly from contextual encoders overall, but this is predominantly due to better labeling of core roles (+19.0 F1 for ELMo) which are known to be closely tied to syntax (e.g. Punyakanok et al. (2008); Gildea & Palmer (2002)). The lexical baseline performs similarly on core and non-core roles (74 and 75 F1 for ELMo), but the more semantically-oriented non-core role

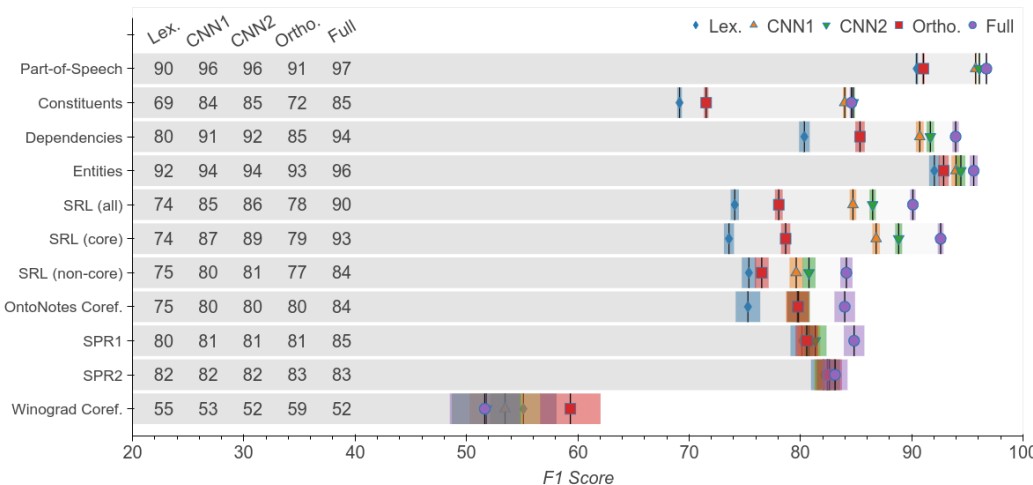

Figure 2: Additional baselines for ELMo, evaluated on the test sets. CNN$k$ adds a convolutional layer that sees $\pm k$ tokens to each side of the center word. Lexical is the lexical baseline, equivalent to $k = 0$. Orthonormal is the full ELMo architecture with random orthonormal LSTM and projection weights, but using the pretrained lexical layer. Full (pretrained) is the full ELMo model. Colored bands are 95% confidence intervals (normal approximation).

labels (such as purpose, cause, or negation) see only a smaller improvement from encoded context (+8.8 F1 for ELMo). The semantic proto-role labeling task (SPR1, SPR2) looks at the same type of core predicate-argument pairs but tests for higher-level semantic properties (§ 2), which we find to be only weakly captured by the contextual encoder (+1-5 F1 for ELMo).

The SemEval relation classification task is designed to require semantic reasoning, but in this case we see a large improvement from contextual encoders, with ELMo improving by 22 F1 points on the lexical baseline (50% relative error reduction) and BERT-large improving by another 4.6 points. We attribute this partly to the poor performance (51-58 F1) of lexical priors on this task, and to the fact that many easy relations can be resolved simply by observing key words in the sentence (for example, "*caused*" suggests the presence of a `Cause-Effect` relation). To test this, we augment the lexical baseline with a bag-of-words feature, and find that for relation classification we capture more than 70% of the headroom from using the full ELMo model.[8]

**Effects of architecture.** Focusing on the ELMo model, we ask: how much of the model's performance can be attributed to the architecture, rather than knowledge from pretraining? In Figure 2 we compare to an orthonormal encoder (§ 4) which is structurally identical to ELMo but contains no information in the recurrent weights. It can be thought of as a randomized feature function over the sentence, and provides a baseline for how the architecture itself can encode useful contextual information. We find that the orthonormal encoder improves significantly on the lexical baseline, but that overall the learned weights account for over 70% of the improvements from full ELMo.

**Encoding non-local context.** How much information is carried over long distances (several tokens or more) in the sentence? To estimate this, we extend our lexical baseline with a convolutional layer, which allows the probing classifier to use local context. In Figure 2 we find that adding a CNN of width 3 ($\pm 1$ token) closes 72% (macro average over tasks) of the gap between the lexical baseline and full ELMo; this extends to 79% if we use a CNN of width 5 ($\pm 2$ tokens). On nonterminal constituents, we find that the CNN $\pm 2$ model matches ELMo performance, suggesting that while the ELMo encoder propagates a large amount of information about constituents (+15.4 F1 vs. Lex., Table 2), most of it is local in nature. We see a similar trend on the other syntactic tasks, with 80-90% of ELMo performance on dependencies, part-of-speech, and SRL core roles captured by CNN $\pm 2$. Conversely, on more semantic tasks, such as coreference, SRL non-core roles, and SPR, the

---

[8]For completeness, we repeat the same experiment on the rest of our task suite. The bag-of-words feature captures 20-50% (depending on encoder) of the full-encoder headroom for entity typing, and much smaller fractions on other tasks.

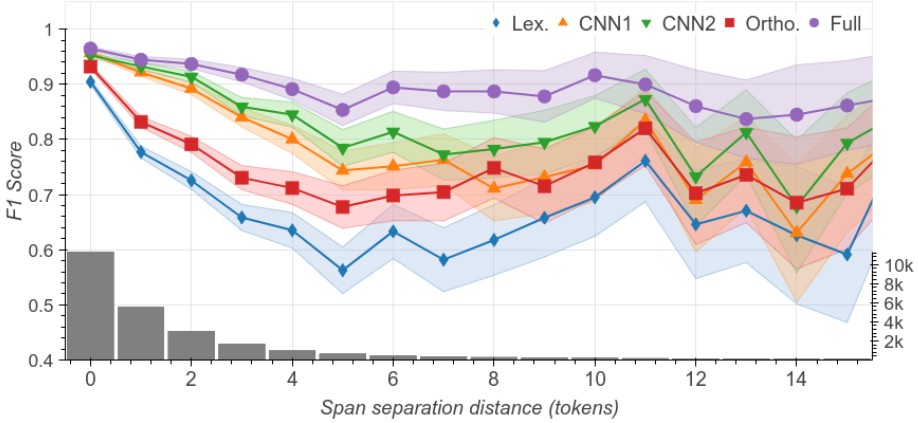

Figure 3: Dependency labeling F1 score as a function of separating distance between the two spans. Distance 0 denotes adjacent tokens. Colored bands are 95% confidence intervals (normal approximation). Bars on the bottom show the number of targets (in the development set) with that distance. Lex., CNN1, CNN2, Ortho, and Full are as in Figure 2.

gap between full ELMo and the CNN baselines is larger. This suggests that while ELMo does not encode these phenomena as efficiently, the improvements it does bring are largely due to long-range information.

We can test this hypothesis by seeing how our probing model performs with distant spans. Figure 3 shows F1 score as a function of the distance (number of tokens) between a token and its head for the dependency labeling task. The CNN models and the orthonormal encoder perform best with nearby spans, but fall off rapidly as token distance increases. The full ELMo model holds up better, with performance dropping only 7 F1 points between $d = 0$ tokens and $d = 8$, suggesting the pretrained encoder does encode useful long-distance dependencies.

## 6 RELATED WORK

Recent work has consistently demonstrated the strong empirical performance of contextualized word representations, including CoVe (McCann et al., 2017), ULMFit (Howard & Ruder, 2018), ELMo (Peters et al., 2018a; Lee et al., 2018; Strubell et al., 2018; Kitaev & Klein, 2018). In response to the impressive results on downstream tasks, a line of work has emerged with the goal of understanding and comparing such pretrained representations. SentEval (Conneau & Kiela, 2018) and GLUE (Wang et al., 2018) offer suites of application-oriented benchmark tasks, such as sentiment analysis or textual entailment, which combine many types of reasoning and provide valuable aggregate metrics which are indicative of practical performance. A parallel effort, to which this work contributes, seeks to understand what is driving (or hindering) performance gains by using "probing tasks," i.e. tasks which attempt to isolate specific phenomena for the purpose of finer-grained analysis rather than application, as discussed below.

Much work has focused on probing fixed-length sentence encoders, such as InferSent (Conneau et al., 2017), specifically their ability to capture surface properties of sentences such as length, word content, and word order (Adi et al., 2017), as well as a broader set of syntactic features, such as tree depth and tense (Conneau et al., 2018). Other related work uses perplexity scores to test whether language models learn to encode properties such as subject-verb agreement (Linzen et al., 2016; Gulordava et al., 2018; Marvin & Linzen, 2018; Kuncoro et al., 2018).

Often, probing tasks take the form of "challenge sets", or test sets which are generated using templates and/or perturbations of existing test sets in order to isolate particular linguistic phenomena, e.g. compositional reasoning (Dasgupta et al., 2018; Ettinger et al., 2018). This approach is exemplified by the recently-released Diverse Natural Language Collection (DNC) (Poliak et al., 2018b), which introduces a suite of 11 tasks targeting different semantic phenomena. In the DNC, these tasks are all recast into natural language inference (NLI) format (White et al., 2017), i.e. systems must understand the targeted semantic phenomenon in order to make correct inferences about en-

tailment. Poliak et al. (2018a) used an earlier version of recast NLI to test NMT encoders' ability to understand coreference, SPR, and paraphrastic inference.

Challenge sets which operate on full sentence encodings introduce confounds into the analysis, since sentence representation models must pool word-level representations over the entire sequence. This makes it difficult to infer whether the relevant information is encoded within the span of interest or rather inferred from diffuse information elsewhere in the sentence. One strategy to control for this is the use of minimally-differing sentence pairs (Poliak et al., 2018b; Ettinger et al., 2018). An alternative approach, which we adopt in this paper, is to directly probe the token representations for word- and phrase-level properties. This approach has been used previously to show that the representations learned by neural machine translation systems encode token-level properties like part-of-speech, semantic tags, and morphology (Shi et al., 2016; Belinkov et al., 2017a;b), as well as pairwise dependency relations (Belinkov, 2018). Blevins et al. (2018) goes further to explore how part-of-speech and hierarchical constituent structure are encoded by different pretraining objectives and at different layers of the model. Peters et al. (2018b) presents similar results for ELMo and architectural variants.

Compared to existing work, we extend sub-sentence probing to a broader range of syntactic and semantic tasks, including long-range and high-level relations such as predicate-argument structure. Our approach can incorporate existing annotated datasets without the need for templated data generation, and admits fine-grained analysis by label and by metadata such as span distance. We note that some of the tasks we explore overlap with those included in the DNC, in particular, named entities, SPR and Winograd. However, our focus on probing token-level representations directly, rather than pooling over the whole sentence, provides a complementary means for analyzing these representations and diagnosing the particular advantages of contextualized vs. conventional word embeddings.

# 7 CONCLUSION

We introduce a suite of "edge probing" tasks designed to probe the sub-sentential structure of contextualized word embeddings. These tasks are derived from core NLP tasks and encompass a range of syntactic and semantic phenomena. We use these tasks to explore how contextual embeddings improve on their lexical (context-independent) baselines. We focus on four recent models for contextualized word embeddings–CoVe, ELMo, OpenAI GPT, and BERT.

Based on our analysis, we find evidence suggesting the following trends. First, in general, contextualized embeddings improve over their non-contextualized counterparts largely on syntactic tasks (e.g. constituent labeling) in comparison to semantic tasks (e.g. coreference), suggesting that these embeddings encode syntax more so than higher-level semantics. Second, the performance of ELMo cannot be fully explained by a model with access to local context, suggesting that the contextualized representations do encode distant linguistic information, which can help disambiguate longer-range dependency relations and higher-level syntactic structures.

We release our data processing and model code, and hope that this can be a useful tool to facilitate understanding of, and improvements in, contextualized word embedding models.

## ACKNOWLEDGMENTS

This work was conducted in part at the 2018 Frederick Jelinek Memorial Summer Workshop on Speech and Language Technologies, and supported by Johns Hopkins University with unrestricted gifts from Amazon, Facebook, Google, Microsoft and Mitsubishi Electric Research Laboratories, as well as a team-specific donation of computing resources from Google. PX, AP, and BVD were supported by DARPA AIDA and LORELEI. Special thanks to Jacob Devlin for providing checkpoints of GPT model trained on the BWB corpus, and to the members of the Google AI Language team for many productive discussions.

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

## A    CHANGES FROM ORIGINAL VERSION

This version of the paper has been updated to include probing results on the popular BERT (Devlin et al., 2018) model, which was released after our original submission. Aside from formatting and minor re-wording, the following changes have been made:

- We include probing results on the BERT-base and BERT-large models (Devlin et al., 2018).
- We add one additional task to Table 2, relation classification on SemEval 2010 Task 8 (Hendrickx et al., 2009), in order to better explore how pre-trained encoders capture semantic information.
- We refer to the OpenAI Transformer LM (Radford et al., 2018) as "GPT" to better reflect common usage.
- We add experiments with ELMo-style scalar mixing (Section 3.2) on the OpenAI GPT model. This improves performance slightly, and changes our conclusion that ELMo was overall superior to GPT; the two are approximately equal on average, with slight differences on some tasks.
- To reduce noise, we report the average over five runs for experiments on Winograd coreference (DPR).

## B    DATASET STATISTICS

Table 3: For each probing task, corpus summary statistics of the number of labels, examples, tokens and targets (split by train/dev/test). Examples generally refer to sentences. For semantic role labeling, they instead refer to the total number of frames. Targets refer to the total number of classification targets (edges or spans, as described in Table 1 and Section 2). For SemEval relation classification there is no standard development split, so we use a fixed subset of 15% of the training data and use the remaining 85% to train.

| Task | $|\mathcal{L}|$ | Examples | Tokens | Total Targets |
|---|---|---|---|---|
| Part-of-Speech | 48 | 116K / 16K / 12K | 2.2M / 305K / 230K | 2.1M / 290K / 212K |
| Constituents | 30 | 116K / 16K / 12K | 2.2M / 305K / 230K | 1.9M / 255K / 191K |
| Dependencies | 49 | 13K / 2.0K / 2.1K | 204K / 25K / 25K | 204K / 25K / 25K |
| Entities | 18 | 116K / 16K / 12K | 2.2M / 305K / 230K | 128K / 20K / 13K |
| SRL (all) | 66 | 253K / 35K / 24K | 6.6M / 934K / 640K | 599K / 83K / 56K |
|     Core roles | 6 | 253K / 35K / 24K | 6.6M / 934K / 640K | 411K / 57K / 38K |
|       Non-core roles | 21 | 253K / 35K / 24K | 6.6M / 934K / 640K | 170K / 24K / 16K |
| OntoNotes coref. | 2 | 116K / 16K / 12K | 2.2M / 305K / 230K | 248K / 43K / 40K |
| SPR1 | 18 | 3.8K / 513 / 551 | 81K / 11K / 12K | 7.6K / 1.1k / 1.1K |
| SPR2 | 20 | 2.2K / 291 / 276 | 47K / 4.9K / 5.6K | 4.9K / 630 / 582 |
| Winograd coref. | 2 | 1.0K / 2.0K / 2.1K | 14K / 8.0K / 14K | 1.8K / 949 / 379 |
| Rel. (SemEval) | 19 | 6.9K / 1.1K / 2.7K | 117K / 20K / 47K | 6.9K / 1.1K / 2.7K |

## C    MODEL DETAILS

Because the vectors have varying dimension across probed models, and to improve performance we first project the vectors down to 256 dimensions:

$$e_i^{(k)} = A^{(k)} e_i + b^{(k)} \tag{1}$$

We use separate projections ($k = 1, 2$) so that the model can extract different information from $s^{(1)}$ (for example, a predicate) and $s^{(2)}$ (for example, an argument). We then apply a pooling operator over the representations within a span to yield a fixed-length representation:

$$r^{(k)}(s_k) = r^{(k)}(i_k, j_k) = \text{Pool}(e_{i_k}^{(k)}, e_{i_k+1}^{(k)}, \ldots, e_{j_k-1}^{(k)}) \tag{2}$$

We use the self-attentional pooling operator from Lee et al. (2017) and He et al. (2018). This learns a weight $z_i^{(k)} = W_{att}^{(k)} e_i^{(k)}$ for each token, then represents the span as a sum of the vectors $e_{i_k}^{(k)}, e_{i_k+1}^{(k)}, \ldots, e_{j_k-1}^{(k)}$ weighted by $a_i^{(k)} = \text{softmax}(\mathbf{z}^{(k)})_i$.

Finally, the pooled span representations are fed into a two-layer MLP followed by a sigmoid output layer:

$$h = MLP([r^{(1)}(s^{(1)}), r^{(2)}(s^{(2)})])$$
$$P(\text{label}_\ell = 1) = \sigma(Wh + b)_\ell \quad \text{for} \quad \ell = 0, \ldots, |\mathcal{L}| \tag{3}$$

We train by minimizing binary cross entropy against the set of true labels. While convention on many tasks (e.g. SRL) is to use a softmax loss, this enforces an exclusivity constraint. By using a per-label sigmoid our model can estimate each label independently, which allows us to stratify our analysis (see § 5) to individual labels or groups of labels within a task.

With the exception of ELMo scalars, we hold the weights of the sentence encoder (§ 3.2) fixed while we train our probing classifier. We train using the Adam optimizer (Kingma & Ba, 2015) with a batch size[9] of 32, an initial learning rate of 1e-4, and gradient clipping with max $L_2$ norm of 5.0. We evaluate on the validation set every 1000 steps (or every 100 for SPR1, SPR2, and Winograd), halve the learning rate if no improvement is seen in 5 validations, and stop training if no improvement is seen in 20 validations.

## D    CONTEXTUAL REPRESENTATION MODELS

**CoVe**    The CoVe model (McCann et al., 2017) is a two-layer biLSTM trained as the encoder side of a sequence-to-sequence(Sutskever et al., 2014) English-German machine translation model. We use the original authors' implementation and the best released pre-trained model [10]. This model is trained on the WMT2017 dataset Bojar et al. (2017) which contains approximately 7 million sentences of English text. Following McCann et al. (2017), we concatenate the activations of the top-layer forward and backward LSTMs ($d = 300$ each) with the pre-trained GloVe (Pennington et al., 2014) embedding[11] ($d = 300$) of each token, for a total representation dimension of $d = 900$.

**ELMo**    The ELMo model (Peters et al., 2018a) is a two layer LSTM trained as the concatenation of a forward and a backward language model, and built over a context-independent character CNN layer. We use the original authors' implementation as provided in the AllenNLP (Gardner et al., 2018) toolkit[12] and the standard pre-trained model trained on the Billion Word Benchmark (BWB) (Chelba et al., 2014)We take the (fixed, contextual) representation of token $i$ to be the set of three vectors $h_{0,i}$, $h_{1,i}$, and $h_{2,i}$ containing the activations of each layer of the ELMo model. Following Equation 1 of Peters et al. (2018a), we learn task-specific scalar parameters and take a weighted sum:

$$e_i = \gamma \left(s_0 h_{0,i} + s_1 h_{1,i} + s_2 h_{2,i}\right) \quad \text{for } i = 0, 1, \ldots, n \tag{4}$$

to give 1024-dimensional representations for each token.

**OpenAI GPT**    The GPT model (Radford et al., 2018) was recently shown to outperform ELMo on a number of downstream tasks, and as of submission holds the highest score on the GLUE benchmark (Wang et al., 2018). It consists of a 12-layer Transformer (Vaswani et al., 2017) model, trained as a left-to-right language model using masked attention. We use a PyTorch reimplementation of

---

[9]For most tasks this is $b = 32$ sentences, which each have a variable number of target spans. For SRL, this is $b = 32$ predicates.

[10]https://github.com/salesforce/cove

[11]glove.840b.300d from https://nlp.stanford.edu/projects/glove/

[12]https://allennlp.org/ version 0.5.1

the model[13], and the pre-trained weights[14] trained on the Toronto Book Corpus (Zhu et al., 2015) Unlike Radford et al. (2018), we hold the Transformer weights fixed while training our probing model in order to better understand what information is available from the pre-training procedure alone. To facilitate more direct comparison with ELMo and CoVe we concatenate (`cat`) the activations of the final Transformer layer ($d = 768$) with the context-independent subword embeddings ($d = 768$) to give contextual vectors of $d = 1536$ for each (sub)-token. We also experiment with ELMo-style scalar mixing (`mix`), which uses additional weight parameters for each layer (embeddings plus layers $1 - 12$) learned for each probing task to give a contextual vector of $d = 768$ for each (sub)-token.

**BERT**   The BERT model of Devlin et al. (2018) has recently shown state-of-the-art performance on a broad set of NLP tasks, outperforming ELMo and the OpenAI Transformer LM. It consists of a stack of Transformer (Vaswani et al., 2017) layers trained jointly as a masked language model and on a next-sentence prediction task. We use a PyTorch reimplementation of the model via the `pytorch_pretrained_bert` package[15], and the pre-trained `bert-base-uncased` (12-layer) and `bert-large-uncased` (24-layer) models trained on the concatenation of the Toronto Books Corpus (Zhu et al., 2015, 800M words of fiction books) and English Wikipedia (2.5B words). Unlike standard usage of the BERT model (Devlin et al., 2018), we hold the Transformer weights fixed while training our probing model. We produce `cat` and `mix` representations with dimensionality $d = 1536$ and $d = 768$, respectively for BERT-base and $d = 2048$ and $d = 1024$ for BERT-large.

## E   RETOKENIZATION

The pre-trained encoder models expect a particular tokenization of the input string, which does not always match the original tokenization of each probing set. To correct this we retokenize the probing data to match the tokenization of each encoder, which for CoVe is Moses tokenization, and for GPT and BERT is a custom subword model (Sennrich et al., 2016; Wu et al., 2016). We then align the spans to the new tokenization using a heuristic projection based on byte-level Levenshtein distance.

The source data for our probing tasks is annotated with respect to a particular tokenization, typically the conventions of the source treebanks (Penn Treebank, Universal Dependencies, and OntoNotes 5.0). This does not always align to the tokenization of the pre-trained representation models. Consider a dummy sentence:

- Text: `I don't like pineapples.`
- Native: `[I do n't like pineapples .]`
- Moses: `[I do n \'t like pineapples .]`
- Subword: `[_i _do _n't _like _pinea pples .]`

An annotation on the word "pineapples" might be expressed as $s = [4, 5)$ in the original ("native") tokenization, but the corresponding text is span $s_{\text{Moses}} = [5, 6)$ under Moses tokenization and $s_{\text{subword}} = [5, 7)$ under the particular subword model above.

We resolve this by aligning the source and target tokenization using Levenshtein distance. We take the source tokenization $[s_0, s_1, \ldots, s_m]$ as given, and treat the target tokenizer as a black-box function from a string $\tilde{S}$ to a list of tokens $[t_0, t_1, \ldots, t_n]$ (note that in general, $n \neq m$). Let $\tilde{S}$ be the source string. We create a target string $\tilde{T}$ by joining $[t_0, t_1, \ldots, t_n]$ with spaces, and then compute a byte-level Levenshtein alignment[16] $\tilde{A} = \text{Align}(\tilde{T}, \tilde{S})$. We then compute token-to-byte alignments $U = \text{Align}([t_0, t_1, \ldots, t_n], \tilde{T})$ and $V = \text{Align}([s_0, s_1, \ldots, s_m], \tilde{S})$. Representing

---

[13]`https://github.com/huggingface/pytorch-openai-transformer-lm`, which we manually verified to produce identical activations to the authors' TensorFlow implementation.

[14]`https://github.com/openai/finetune-transformer-lm`

[15]`https://github.com/huggingface/pytorch-pretrained-BERT` version 0.4.0, which has been verified to reproduce the activations of the original TensorFlow implementation.

[16]We use the `python-Levenshtein` package, `https://pypi.org/project/python-Levenshtein/`

the alignments as boolean adjacency matricies, we can compose them to form a token-to-token alignment $A = U\tilde{A}V^T$.

We then represent each source span as a boolean vector with 1s inside the span and 0s outside, e.g. $[2, 4) = [0, 0, 1, 1, 0, 0, \ldots] \in \{0, 1\}^m$, and project through the alignment $A$ to the target side. We recover a target-side span from the minimum and maximum nonzero indices.

