# OpenReview forum: "What do you learn from context? Probing for sentence structure in contextualized word representations"
_ICLR.cc/2019/Conference_

### Official Review · AnonReviewer3 · 2018-11-01
**Current work reps capture a surprising amount of structure**

**Rating:** 7
**Confidence:** 4

**Review:**

I have no major complaints with this work.  It is well presented and easily understandable. I agree with the claim that the largest gains are largely syntactic, but this leads me to wonder about more tail phenomena.   PP attachment is a classic example of a syntactic decision requiring semantics, but one could also imagine doing a CCG supertagging analysis to see how well the model captures specific long-tail phenomena.  Though a very different task Vaswani et al 16, for example, showed how bi-LSTMs were necessary for certain constructions (presumably current models would perform much better and may capture this information already).

An important caveat of these results is that the evaluation (by necessity) is occurring in English.  Discourse in a pro-drop language would presumably require longer contexts than many of these approaches currently handle.

---

> ### Author Response · Authors · 2018-11-14
> **Author response**
>
> Thank you for the review!
>
> We agree that it would be interesting to explore more specific tail phenomena. Attachment phenomena in particular can be studied on many of the same datasets if we fix labels and instead predict one of the two spans; this would be an interesting direction for future study.
>
> It would also be very interesting to explore other languages! While we are limited by available data and encoder models, there’s nothing in the edge probing technique that makes English-specific assumptions. Probing for phenomena that require long contexts could be a good test of advanced encoders, and can be easily quantified in our framework (for example, see Figure 3).

---

### Official Review · AnonReviewer2 · 2018-11-02
**Nice discussion of what type of information is actually encoded by contextualized word embeddings**

**Rating:** 7
**Confidence:** 4

**Review:**

This paper provides new insights on what is captured contextualized word embeddings by compiling a set of “edge probing” tasks.  This is not the first paper to attempt this type of analysis, but the results seem pretty thorough and cover a wider range of tasks than some similar previous works.  The findings in this paper are very timely and relevant given the increasing usage of these types of embeddings.  I imagine that the edge probing tasks could be extended towards looking for other linguistic attributes getting encoded in these embeddings.

Questions & other remarks:
-The discussion of the tables and graphs in the running text feels a bit condensed and at times unclear about which rows are being referred to.
-In figures 2 & 3: what are the tinted areas around the lines signifying here? Standard deviation?  Standard error?  Confidence intervals?
-It seems the orthonormal encoder actually outperforms the full elmo model with the learned weights on the Winograd Schema.  Can the authors comment on this a bit more?

---

> ### Author Response · Authors · 2018-11-14
> **Author response**
>
> Thank you for the review!
>
> We’re very interested in probing for other linguistic attributes - while we present a broad analysis in this paper, there’s certainly room to use edge probing to study more focused phenomena like PP attachment or ambiguities between specific semantic roles. We use a standardized data format that makes it easy to add new tasks, and we hope that our code release will be a useful platform for this kind of analysis.
>
> We’ll be sure to update the text to more clearly describe the tables.
>
> Whoops! In Figure 2 and 3, the bars/bands are 95% confidence intervals calculated using the Normal approximation. We wanted to emphasize that the SPR and Winograd datasets are quite small and that the differences between models are often not significant. We’ll add this to the caption in the final version.

---

### Official Review · AnonReviewer1 · 2018-11-06
**Nice empirical paper**

**Rating:** 7
**Confidence:** 4

**Review:**


This is a nice paper that attempts to tease apart some questions about the effectiveness of contextual word embeddings (ELMo, CoVe, and the Transformer LM). The main question is about the value of context in these representations, and in particular how their ability to encode context allows them to also (implicitly) represent linguistic properties of words. What I really like about the paper is the “Edge probing” method it introduces. The idea is to probe the representations using diagnostic classifiers—something that’s already widespread practice—but to focus on the relationship between spans rather than individual words. This is really nice because it enables them to look at more than just tagging problems: the paper looks at syntactic constituency, dependencies, entity labels, and semantic role labeling. I think the combination of an interesting research question and a new method (which will probably be picked up by others working in this area) make this a strong candidate for ICLR. The paper is well-written and experimentally thorough.

Nitpick: It would be nice to see some examples of cases where the edge probe is correct, and where it isn’t.

---

> ### Author Response · Authors · 2018-11-14
> **Author response**
>
> Thank you for the review! We do hope that this will be of broad interest given recent progress in sentence representations, and hope that our code release will allow continued evaluation of new and better representation models (like BERT).
>
> We’ll certainly include examples of specific win / loss cases in the final version.

---

### Public Comment · (anonymous) · 2018-11-19
**Some previous work on edge probing**

Just wanted to mention a related work: Yonatan Belinkov's thesis ( http://people.csail.mit.edu/belinkov/assets/pdf/thesis2018.pdf ) has some prior experiments with the edge probing task design outlined in this paper. See Chapter 4, "Sentence Structure and Neural Machine Translation: Word Relations".

---

> ### Author Response · Authors · 2018-11-26
> **Thanks for the reference**
>
> This is definitely related; we'll be sure to add a citation!

---

### Meta-Review · Area_Chair1 · 2018-12-13
**A thorough study of contextualized word representations**

**Confidence:** 5
**Recommendation:** Accept (Poster)

**Metareview:**

Pros

- Thorough analysis on a large number of diverse tasks
- Extending the probing technique typically applied to individual encoder states to testing for presence of certain (linguistic) information based on pairs of encoders states (corresponding to pairs of words)
- The comparison can be useful when deciding which representations to use for a given task

Cons

- Nothing serious, it is solid and important empirical study

The reviewers are in consensus.